# Unraveling the Complexity of Memory in RL Agents: an Approach for Classification and Evaluation

## Abstract

The incorporation of memory into agents is essential for numerous tasks within the domain of Reinforcement Learning (RL). In particular, memory is paramount for tasks that require the utilization of past information, adaptation to novel environments, and improved sample efficiency. However, the term "memory" encompasses a wide range of concepts, which, coupled with the lack of a unified methodology for validating an agent's memory, leads to erroneous judgments about agents' memory capabilities and prevents objective comparison with other memory-enhanced agents. This paper aims to streamline the concept of memory by providing precise definitions of agent memory types, such as long-term versus short-term memory and declarative versus procedural memory, inspired by cognitive science. Using these definitions, we categorize different classes of agent memory, propose a robust experimental methodology for evaluating the memory capabilities of RL agents, and standardize evaluations. Furthermore, we empirically demonstrate the importance of adhering to the proposed methodology when evaluating different types of agent memory by conducting experiments with different RL agents and what its violation leads to.

## 1 Introduction

Reinforcement Learning (RL) effectively addresses various problems within the Markov Decision Process (MDP) framework, where agents make decisions based on immediately available information (Mnih et al., 2015; Badia et al., 2020). However, there are still challenges in applying RL to more complex tasks with partial observability.

To successfully address such challenges, it is essential that an agent is able to efficiently store and process the history of its interactions with the environment (Ni et al., 2021). Sequence processing methods originally developed for natural language processing (NLP) can be effectively applied to these tasks because the history of interactions with the environment can be represented as a sequence (Hausknecht & Stone, 2015; Esslinger et al., 2022; Samsami et al., 2024).

However, in many tasks, due to the complexity or noisiness of observations, the sparsity of events, the difficulty of designing the reward function, and the long duration of episodes, storing and retrieving important information becomes extremely challenging, and the need for memory mechanisms arises (Graves et al., 2016; Wayne et al., 2018; Goyal et al., 2022). Nevertheless, in the existing literature on RL, where the concept of "memory" is discussed, the definitions of memory are only defined in terms of the specific problem under consideration.

For example, in some works, memory is defined as the ability of an agent to effectively establish and use dependencies between events within a fixed-size sequence of tokens (context) in decision making (Esslinger et al., 2022; Ni et al., 2023; Grigsby et al., 2024). In other works, the term "memory" refers to the agent's ability to use out-of-context information through the use of various memory mechanisms (Parisotto et al., 2020; Lampinen et al., 2021; Cherepanov et al., 2024). In the context of Meta-Reinforcement Learning (Meta-RL), however, the term "memory" is used to describe an agent's ability to use experience from other tasks or episodes to adapt to a new, previously unknown environment (Team et al., 2023; Kang et al., 2024a; Grigsby et al., 2024).

To avoid possible cases of ambiguous interpretation of the concept of memory in RL and, as a consequence, the conduct of incorrect experiments, we introduce a formal definition of agent memory in RL, distinguish classes of tasks that require an agent to have memory, and introduce a classification of agent memory types. Furthermore, we propose a formal methodology that can be used to unambiguously test different agent memory capabilities according to our proposed classification.

In summary, our contribution can be described as follows:

1. We formalize the definition of "memory" depending on the problem to be solved: *long-term memory* and *short-term memory*, *declarative memory* and *procedural memory* (section 5).

2. We introduce a decoupling of tasks that require an agent to have memory into two classes: *Memory Decision-Making* (*Memory DM*) and *Meta-Reinforcement Learning* (*Meta-RL*) (section 5).

3. We propose a generic experimental methodology for testing the memory capabilities of agents in Memory DM tasks (subsection 5.2).

4. We show that if the proposed experimental methodology is not followed, judgments about the agent's memory type can become extremely incorrect (section 6).

## 2 PARTIALLY OBSERVABLE MARKOV DECISION PROCESS

The Partially Observable Markov Decision Process (POMDP) is a generalization of the Markov Decision Process (MDP) that models sequential decision-making problems where the agent has incomplete information about the environment's state. POMDP can be represented as a tuple $\mathcal{M}_P = \langle \mathcal{S}, \mathcal{A}, \mathcal{O}, \mathcal{P}, \mathcal{R}, \mathcal{Z} \rangle$, where $\mathcal{S}$ denotes the set of states, $\mathcal{A}$ is the set of actions, $\mathcal{O}$ is the set of observations and $\mathcal{Z} = \mathcal{P}(o_{t+1} \mid s_{t+1}, a_t)$ is an observation function such that $o_{t+1} \sim \mathcal{Z}(s_{t+1}, a_t)$. An agent takes an action $a_t \in \mathcal{A}$ based on the observed history $h_{0:t-1} = \{(o_i, a_i, r_i)\}_{i=0}^{t-1}$ and receives a reward $r_t = \mathcal{R}(s_t, a_t)$. It is important to note that state $s_t$ is not available to the agent at time $t$. In the case of POMDPs, a policy is a function $\pi(a_t \mid o_t, h_{0:t-1})$ that uses the agent history $h_{0:t-1}$ to obtain the probability of the action $a_t$. Thus, in order to operate effectively in a POMDPs, an agent must have memory mechanisms to retrieve a history $h_{0:t-1}$. Partial observability arises in a variety of real-world situations, including robotic navigation and manipulation tasks, autonomous vehicle tasks, and complex decision-making problems.

## 3 RELATED WORKS

Researchers' interest in memory-enhanced RL agents is evident in the abundance of works proposing architectures with memory mechanisms and benchmarks (Osband et al., 2019; Morad et al., 2023; Pleines et al., 2023) for their validation (see Appendix B and Appendix D for details). However, despite the rather large number of works devoted to this topic, the term "memory" has multiple senses, and the selection of benchmarks and experiments is not always done correctly.

Thus, for instance, in Oh et al. (2016), memory is understood as the ability of an agent to store recent observations into an external buffer and then retrieve relevant information based on temporal context. In Lampinen et al. (2021), memory is the ability to store and recall desired information at long intervals. In Fortunato et al. (2020), memory refers to working and episodic memory (with short-term and long-term nature, respectively) from cognitive psychology and neuroscience, which allows an intelligent agent to use information from past events to make decisions in the present and future. Ni et al. (2023) describes two distinct forms of temporal reasoning: (working) memory and (temporal) credit assignment, where memory refers to the ability to recall a distant past event at the current time. In Kang et al. (2024b) authors use the concept of reconstructive memory Bartlett & Kintsch (1995) discovered in psychology, which establishes a reflection process based on interaction.

## 4 MEMORY OF HUMANS AND AGENTS

Most works related to the concept of memory in RL use various principles from cognitive psychology and neuroscience such as long-term memory (Lampinen et al., 2021; Ni et al., 2023; Grigsby et al., 2024), working memory (Graves et al., 2014; Fortunato et al., 2020), episodic memory (Pritzel et al.,

2017; Fortunato et al., 2020), associative memory (Parisotto & Salakhutdinov, 2017; Zhu et al., 2020), and others to introduce it. Despite the fundamental differences in these concepts, works on memory in RL often simplify these concepts to their inherent temporal scales (short-term memory and long-term memory). Regardless, the temporal scales are often presented qualitatively without clearly defining the boundaries between them. For example, many studies assume that remembering a few steps within an environment represents short-term memory, while remembering hundreds of steps represents long-term memory, without considering the relative nature of these concepts. This ambiguity between short-term and long-term memory can lead to a misattribution of an agent's memory capabilities and to an incorrect estimation of them when conducting experiments. To address this ambiguity, in this section we introduce formal definitions of agent memory in RL and its types, and propose an algorithm for designing an experiment to test agent memory in a correct way.

## 4.1 MEMORY IN COGNITIVE SCIENCE

Human cognitive abilities that ensure adaptive survival depend largely on memory, which determines the accumulation, preservation, and reproduction of knowledge and skills (Parr et al., 2020; 2022). Memory exists in many forms, each of which relies on different neural mechanisms. Neuroscience and cognitive psychology distinguish memory by the temporal scales at which information is stored and accessed, and by the type of information that is stored. Abstracting from this distinction, a high-level definition of human memory is as follows: "***memory – is the ability to retain information and recall it at a later time***".

The definition aligns with the common understanding of memory in RL. Thus, we will use it to create terminology for various types of agent memory. In neuroscience, memory is categorized by temporal scale and behavioral manifestation. Typically, this leads to a distinction between *short-term memory*, which retains information for seconds, and *long-term memory*, which can last a lifetime (Davis & Squire, 1984). Additionally, memory is divided by behavioral manifestations into *declarative memory* (explicit) and *procedural memory* (implicit) (Graf & Schacter, 1985). Declarative memories can be consciously recalled, encompassing events and facts, while procedural memories are unconscious and relate to skills like skiing or driving.

In the following section, we introduce formal definitions of the above types of memory from neuroscience for RL tasks. Using these definitions, which are written in quantitative terms, we can uniquely classify the type of memory an agent has when using past information in decision making.

## 4.2 MEMORY IN RL

The interpretation of memory in RL varies across studies. In some POMDPs, agents need to retain crucial information to make future decisions within a single environment. Here, memory typically encompasses two aspects: 1) the efficiency of establishing dependencies between events within a fixed time interval (e.g., transformer context (Esslinger et al., 2022; Ni et al., 2023)); and 2) the efficiency of establishing dependencies between events outside a fixed time interval (Parisotto et al., 2020; Cherepanov et al., 2024).

Based on the neuroscience definitions outlined in subsection 4.1, the first interpretation aligns with short-term memory, while the second corresponds to long-term memory. Both interpretations are also closely related to declarative memory. In Meta-RL, memory typically refers to an agent's ability to leverage skills from different environments/episodes Team et al. (2023); Kang et al. (2024a), akin to procedural memory.

However, many studies fail to differentiate between agents with declarative and procedural memory, often treating Meta-RL tasks as a whole rather than focusing on decision-making based on past information. For instance, when a paper asserts that an agent possesses long-term memory, it may only be tested on Meta-RL tasks based on MDPs. To clarify the concept of agent memory in RL, we provide formal definitions in this section.

In this paper, we primarily study an agent's memory, which is used to make current decisions based on past information within the same environment. Accordingly, our focus will be on declarative memory, specifically its short-term and long-term forms.

**Memory and Credit Assignment.** Papers exploring agent memory, particularly declarative memory, often distinguish between two concepts based on the temporal dependencies the agent must handle: *memory* and *credit assignment* (Osband et al., 2019; Mesnard et al., 2020; Ni et al., 2023). In Ni et al. (2023), the authors formally differentiate between two forms of temporal reasoning in RL: (working) memory and (temporal) credit assignment: "*memory* refers to the ability to recall a distant past event at the current time, while *credit assignment* refers to the ability to determine when the actions that merit current credit occurred" (Ni et al., 2023).

While distinct, these concepts both establish different temporal dependencies between related events. In this work, we focus on the agent's ability to form these dependencies, treating "memory" and "credit assignment" as a single entity. We will use the definition from subsection 4.1 to define memory generally. Notably, the definitions for "memory" also apply to "credit assignment", as they pertain solely to temporal dependencies rather than their essence.

## 5 MEMORY DECISION MAKING

POMDP tasks that use agent memory can be divided into two main classes: *Meta Reinforcement Learning* (*Meta-RL*), which involves skill transfer across tasks, and *Memory Decision-Making* (*Memory DM*), which focuses on storing and retrieving information for future decisions.

This distinction is crucial: agents in Meta-RL use something like the procedural memory of subsection 4.1 to facilitate rapid learning and generalization, while those in Memory DM rely on something like declarative memory for current decision-making within the same environment. Despite these differences, many studies overlook behavioral manifestations and focus solely on temporal scales.

To introduce a definition for Memory DM tasks, we first need to introduce the definition of agent context length:

**Definition 1. Agent context length** ($K \in \mathbb{N}$) – is the maximum number of previous steps (triplets of $(o, a, r)$) that the agent can process at time $t$.

For example, an MLP-based agent processes one step at a time ($K = 1$), while a transformer-based agent can process a sequence of up to $K = K_{attn}$ triplets, where $K_{attn}$ is determined by attention. Using the introduced Definition 1 for agent context length, we can introduce a formal definition for the Memory DM framework we focus on in this paper:

**Definition 2. Memory Decision-Making** (**Memory DM**) – is a class of POMDPs in which the agents decision-making process at time $t$ is based on the history $h_{0:t-1} = \{(o_i, a_i, r_i)\}_{i=0}^{t-1}$ if $t > 0$ otherwise $h = \varnothing$. The objective is to determine an optimal policy $\pi^*(a_t \mid o_t, h_{0:t-1})$ that maps the current observation $o_t$ and history $h_{0:t-1}$ of length $t$ to an action $a_t$, maximizing the expected cumulative reward within a single POMDP environment $\mathcal{M}_P$: $J^\pi = \mathbb{E}_\pi \left[ \sum_{t=0}^{T-1} \gamma^t r_t \right]$, where $T$ – episode duration, $\gamma \in [0, 1]$ – discount factor.

In the Memory DM framework (Definition 2), memory refers to the agent's ability to recall information from the past within a single environment and episode. In contrast, in the Meta-RL framework (see Appendix, Definition 7), memory involves recalling information about the agent's behavior from other environments or previous episodes. To distinguish these concepts, we adopt the definitions of *"Declarative memory"* and *"Procedural memory"* from subsection 4.1:

**Definition 3** (**Declarative and Procedural memory in RL**). Let $n_{envs}$ be the number of training environments and $n_{eps}$ the number of episodes per environment. Then,

1. **Declarative Memory** – a type of agent memory when an agent transfers its knowledge within a single environment and across a single episode within that environment:

$$\text{Declarative Memory} \iff n_{envs} \times n_{eps} = 1 \tag{1}$$

2. **Procedural Memory** – a type of agent memory when an agent transfers its skills across multiple environments or multiple episodes within a single environment:

$$\text{Procedural Memory} \iff n_{envs} \times n_{eps} > 1 \tag{2}$$

Here, "knowledge" refers to observable information like facts, locations, and events. In contrast, "skills" are pre-learned policies that an agent can apply across various tasks. Thus, the Memory DM framework validates the agent's declarative memory, while the Meta-RL framework validates its procedural memory (see Figure 1).

In subsection 4.2, we distinguished two classes of POMDPs: Memory DM, which requires declarative memory, and Meta-RL, which requires procedural memory. Within the Memory DM tasks, which are our primary focus, agent memory is categorized into long-term memory and short-term memory:

**Definition 4** (**Memory DM types of memory**). Let $K$ be the agent context length, $\alpha_{t_e}^{\Delta t} = \{o_i, a_i, r_i\}_{i=t_e}^{t_e+\Delta t}$ – an event of duration $\Delta t$ that begins at $t = t_e$ and ends at $t = t_e + \Delta t$, and $\beta_{t_r}(\alpha_{t_e}^{\Delta t}) = a_t \mid (o_t, \alpha_{t_e}^{\Delta t})$ – a decision-making point (recall) at time $t = t_r$ based on the current observation $o_t$ and information about the event $\alpha_{t_e}^{\Delta t}$. Let also $\xi = t_r - t_e - \Delta t + 1$ be the *correlation horizon*, i.e. the minimal time delay between the event $\alpha_{t_e}^{\Delta t}$ that supports the decision-making and the moment of recall of this event $\beta_{t_r}$. Then,

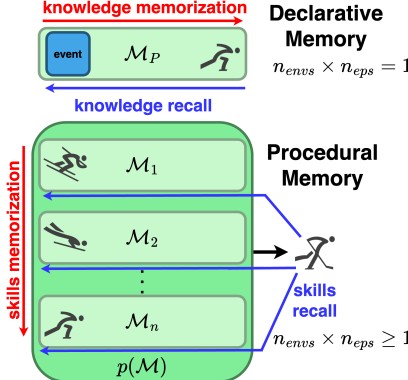

Figure 1: Declarative and procedural memory scheme. Red arrows show the information transfer for memorization, blue arrows show the direction of recall to the required information.

1. **Short-term memory (STM)** – an agent ability to utilize information about local correlations from the past within the agent context of length $K$ at the time of decision making:

$$\text{Short-term memory} \iff \beta_{t_r}(\alpha_{t_e}^{\Delta t}) = a_t \mid (o_t, \alpha_{t_e}^{\Delta t}) \, \forall \, \xi = t_r - t_e - \Delta t + 1 \leq K$$

2. **Long-term memory (LTM)** – an agent ability to utilize information about global correlations from the past outside of the agent context of length $K$, during decision-making:

$$\text{Long-term memory} \iff \beta_{t_r}(\alpha_{t_e}^{\Delta t}) = a_t \mid (o_t, \alpha_{t_e}^{\Delta t}) \, \forall \, \xi = t_r - t_e - \Delta t + 1 > K$$

An illustration for the definitions of classifying Memory DM tasks into LTM and STM from Definition 4 is shown in Figure 2.

The two definitions of declarative memory encompass all work related to Memory DM tasks, where decisions are based on past information. Meta-RL consists of an inner-loop, where the agent interacts with the environment $\mathcal{M} \sim p(\mathcal{M})$, and an outer-loop for transferring knowledge between tasks. Typically, $\mathcal{M}$ is an MDP that doesn't require memory, serving only the outer-loop, which is what "memory" refers to in Meta-RL studies.

The tasks in which the agent makes decisions based on interaction histories in the inner-loop are not named separately, since the classification of Meta-RL task types (multi-task, multi-task 0-shot, and single-task) is based solely on outer-loop parameters ($n_{envs}$ and $n_{eps}$) and does not consider inner-loop task types. However, we can classify the agent's memory for these tasks as declarative short-term or long-term memory (see Figure 3).

We introduce an additional decoupling of Meta-RL task types into green (with POMDP inner-loop tasks) and blue (with MDP inner-loop tasks). In the green case, the agent's memory is required for both skill transfer in the outer-loop

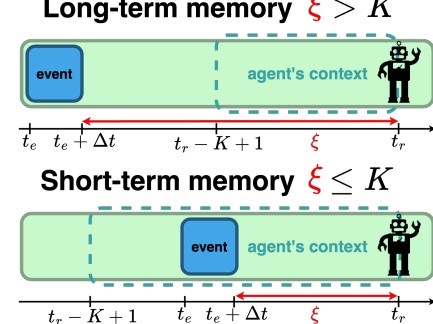

Figure 2: Long-term memory and short-term memory scheme. $t_e$ – event used for decision-making start time, $\Delta t$ – event duration, $t_r$ – agent's recall time, $K$ – agent's context length, $\xi$ – correlation horizon. If an event is outside the context, long-term memory is needed for decision-making; if within the context, short-term memory suffices.

and decision-making based on interaction histories in the inner-loop, and therefore within the inner-loop can be considered as a Memory DM. In the blue case, memory is needed only for skill transfer. While this paper focuses on Memory DM tasks, the terminology allows for further classification of

various Meta-RL tasks, with POMDP sub-classes highlighted in green. The proposed classification of tasks requiring agent memory is presented in Table 1.

Table 1: Classification of tasks requiring agent memory based on our definitions: green indicates tasks described by the proposed definitions, while blue indicates those that are not. Meta-RL tasks with a POMDP inner-loop are marked green as they can be classified as Memory DM tasks. POMDP$^{\dagger}$ indicates a Memory DM task considered as an inner-loop task without an outer-loop.

| Envs. num. | Runs num. | POMDP | Inner-loop task | Memory | Tasks that require agent memory | |
|---|---|---|---|---|---|---|
| | | | | | **Memory DM** | |
| | | | | | Long-term memory ($\xi > K$) | Short-term memory ($\xi \le K$) |
| $n_{envs} = 1$ | $n_{eps} = 1$ | Memory DM | POMDP$^{\dagger}$ | Declarative | Long-term memory task | Short-term memory task |
| | | | | | **Meta-RL: Outer-loop and inner-loop memory** | |
| | | | | | Long-term memory ($\xi > K$) | Short-term memory ($\xi \le K$) |
| $n_{envs} = 1$ | $n_{eps} > 1$ | Meta-RL | POMDP | Procedural | Single-task Meta-RL | Single-task Meta-RL |
| $n_{envs} > 1$ | $n_{eps} = 1$ | Meta-RL | POMDP | Procedural | Multi-task 0-shot Meta-RL | Multi-task 0-shot Meta-RL |
| $n_{envs} > 1$ | $n_{eps} > 1$ | Meta-RL | POMDP | Procedural | Multi-task Meta-RL | Multi-task Meta-RL |
| | | | | | **Meta-RL: Outer-loop memory only** | |
| | | | | | No memory ($\xi = 1$) | No memory ($\xi = 1$) |
| $n_{envs} = 1$ | $n_{eps} > 1$ | Meta-RL | MDP | Procedural | Single-task Meta-RL | Single-task Meta-RL |
| $n_{envs} > 1$ | $n_{eps} = 1$ | Meta-RL | MDP | Procedural | Multi-task 0-shot Meta-RL | Multi-task 0-shot Meta-RL |
| $n_{envs} > 1$ | $n_{eps} > 1$ | Meta-RL | MDP | Procedural | Multi-task Meta-RL | Multi-task Meta-RL |

## 5.1 MEMORY-INTENSIVE ENVIRONMENTS

To effectively test a Memory DM agent's use of short-term and long-term memory, it is crucial to design appropriate experiments. Not all environments are suitable for assessing agent memory; for example, Atari games (Bellemare et al., 2013) with frame stacking or MuJoCo control tasks (Fu et al., 2021) may yield unrepresentative results. To facilitate the evaluation of agent memory capabilities, we formalize the definition of memory-intensive environments:

**Definition 5 (Memory-intensive environments).** Let $\mathcal{M}_P$ be POMDP and $\Xi = \{\xi_n\} = \{(t_r - t_e - \Delta t + 1)_n\}_n$ – set of correlation horizons $\xi$ between for all event-recall pairs. Then $\tilde{\mathcal{M}}_P$ – *memory-intensive* environment $\iff \min_n \Xi > 1$.

**Corollary**: $\max_n \Xi = 1 \iff \mathcal{M} - \text{MDP}$.

Using the definitions of memory-intensive environments (Definition 5) and agent memory types (Definition 4), we can configure experiments to test short-term and long-term memory in the Memory DM framework. Notably, the same memory-intensive environment can validate both types of memory, as outlined in Theorem 1:

**Theorem 1 (On the context memory border).** *Let $\tilde{\mathcal{M}}_P$ be a memory-intensive environment and $K$ be an agents context length. Then there exists context memory border $\overline{K} \ge 1$ such that if $K \le \overline{K}$ then the environment $\tilde{\mathcal{M}}_P$ is used to validate exclusively long-term memory in Memory DM framework:*

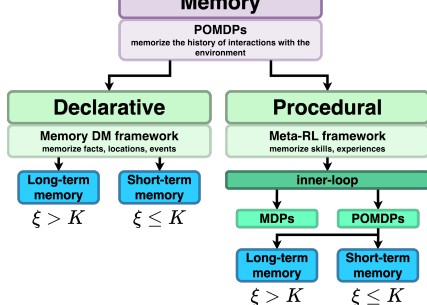

Figure 3: Classification of memory types of RL agents. While the Memory DM framework contrasts with Meta-RL, its formalism can also describe inner-loop tasks when they are POMDPs.

$$\exists \, \overline{K} \ge 1 : \forall \, K \in [1, \overline{K}] : K < \min_n \Xi \tag{3}$$

*Proof.* Let $\overline{K} = \min \Xi - 1$. Then $\forall \, K \le \overline{K}$ is guaranteed that no correlation horizon $\xi$ is in the agent history $h_{t-K+1:t}$, hence the context length $K \le \min \Xi - 1$ generates the long-term memory problem exclusively. Since context length cannot be negative or zero, it turns out that $1 \le K \le \overline{K} = \min \Xi - 1$, which was required to prove. ∎

According to Theorem 1, in a memory-intensive environment $\tilde{\mathcal{M}}_P$, the value of the context memory border $\overline{K}$ can be found as

$$\overline{K} = \min \Xi - 1 = \min_n \left\{ (t_r - t_e - \Delta t + 1)_n \right\}_n - 1 \tag{4}$$

Using Theorem 1, we can establish the necessary conditions for validating short-term memory:

**Weak condition to validate short-term memory**: if $\overline{K} < K < \max \Xi$, then the environment $\tilde{M}_P$ is used to validate both short-term and long-term memory.

**Strong condition to validate short-term memory**: if $\max \Xi < K$, then the environment $\tilde{M}_P$ is used to validate exclusively short-term memory.

According to Theorem 1, if $K \in [1, \overline{K}]$, none of the correlation horizons $\xi$ will be in the agent's context, validating only long-term memory. When $\overline{K} < K < \max \Xi \leq T - 1$, long-term memory can still be tested, but some correlation horizons $\xi$ will fall within the agent's context and won't be used for long-term memory validation. In such a case it is not possible to estimate long-term memory explicitly. When $K \geq \max \Xi$, all correlation horizons $\xi$ are within the agent's context, validating only short-term memory. Summarizing the obtained results, the final division of the required agent context lengths for short-term memory and long-term memory validation is as follows:

1. $K \in [1, \overline{K}] \Rightarrow$ validating long-term memory only.

2. $K \in (\overline{K}, \max \Xi) \Rightarrow$ validating both short-term memory and long-term memory.

3. $K \in [\max \Xi, \infty) \Rightarrow$ validating short-term memory only.

## 5.2 Long-term memory in Memory DM

As defined in Definition 4, Memory DM tasks with short-term memory occur when event-recall pairs in the memory-intensive environment $\tilde{\mathcal{M}}_P$ are within the agent's context ($\xi \leq K$). Here, memory involves the agent's ability to connect information within a context, regardless of how large $K$ is. Examples include works like Esslinger et al. (2022); Ni et al. (2023); Grigsby et al. (2024). Validating short-term memory is straightforward by simply setting a sufficiently large context length $K$. However, validating long-term memory capabilities is more complex and of greater interest.

Memory DM tasks requiring long-term memory occur when event-recall pairs in the memory-intensive environment $\tilde{\mathcal{M}}_P$ are outside the agent's context ($\xi > K$). In this case, memory involves the agent's ability to connect information beyond its context, necessitating memory mechanisms (Definition 6) that can manage interaction histories $h$ longer than the agent's base model can handle.

**Definition 6** (**Memory mechanisms**). Let the agent process histories $h_{t-K+1:t}$ of length $K$ at the current time $t$, where $K \in \mathbb{N}$ is agents context length. Then, a **memory mechanism** $\mu(K) : \mathbb{N} \to \mathbb{N}$ is defined as a function that, for a fixed $K$, allows the agent to process sequences of length $K_{eff} \geq K$, i.e., to establish global correlations out of context, where $K_{eff}$ is the *effective context*.

$$\mu(K) = K_{eff} \geq K \tag{5}$$

Memory mechanisms are essential for addressing long-term memory challenges (processing out-of-context information) in the Memory DM framework.

**Example of memory mechanism.** Consider an agent based on an RNN architecture that can process $K = 1$ triplets of tokens (observations, actions, and rewards) at all times $t$. By using memory mechanisms $\mu(K)$ (e.g., as in Hausknecht & Stone (2015)), the agent can increase the number of tokens processed in a single step without expanding the context size of its RNN architecture. Therefore, if initially in a memory-intensive environment $\tilde{\mathcal{M}}_P : \xi > K = 1$, it can now be represented as $\tilde{\mathcal{M}}_P : \xi \leq K_{eff} = \mu(K)$. Here, the memory mechanism $\mu(K)$ refers to the RNNs recurrent updates to its hidden state.

Thus, validating an agent's ability to solve long-term memory problems in the Memory DM framework reduces to validating the agent's memory mechanisms $\mu(K)$. To design correct experiments in such a case, the following condition must be met:

$$\tilde{\mathcal{M}}_P : K \leq \overline{K} < \xi \leq K_{eff} = \mu(K) \tag{6}$$

According to our definitions, agents with memory mechanisms within the Memory DM framework that can solve long-term memory tasks can also handle short-term memory tasks, but not vice versa. The algorithm for setting up experiments to test an agent's short-term or long-term memory is outlined in Algorithm 1.

---

**Algorithm 1** Algorithm for setting up an experiment to test long-term and short-term memory in Memory DM framework.

---

**Require:** $\tilde{\mathcal{M}}_P$ – memory-intensive environment; $\mu(K)$ – memory mechanism.

**1. Estimate the number of $n$ event-recall pairs in the environment (Definition 5).**

    1. $n = 0 \rightarrow$ Environment is not suitable for testing long-term and short-term memory.

    2. $n \geq 1 \rightarrow$ Environment is suitable for testing long-term and short-term memory.

**2. Estimate context memory border $\overline{K}$ (Equation 4).**

    1. $\forall$ event-recall pair $(\beta(\alpha), \alpha)_i$ find corresponding $\xi_i, i \in [1..n]$.

    2. Determine $\overline{K}$ as $\overline{K} = \min \Xi - 1 = \min_n \{\xi_n\}_n - 1 = \min_n \left\{ (t_r - t_e - \Delta t + 1)_n \right\}_n - 1$

**3. Conduct an appropriate experiment (Definition 4).**

    1. To test short-term memory set $K > \overline{K}$.

    2. To test long-term memory set $K \leq \overline{K} \leq K_{eff} = \mu(K)$.

**4. Analyze the results.**

---

### 5.3 EXAMPLES OF SETTING UP AN EXPERIMENT TO TEST MEMORY IN MEMORY DM FRAMEWORK

**Passive T-Maze.** Consider the Passive T-Maze environment (Ni et al., 2023), where the agent starts at the beginning of a T-shaped corridor and observes a clue that is only available at that location. To complete the episode, the agent must walk straight to the junction and turn based on the initial clue. This environment is defined by the corridor length $L$, with episode duration $T = L + 1$. We will analyze this environment using the Algorithm 1:

1. There is only one event-recall pair in the environment (observing a clue – turning at the junction), so $n = 1$, making it suitable for testing both long-term and short-term memory.

2. The duration of this event is $\Delta t = 0$ (the clue available only at one timestep), and the correlation horizon $\xi = T - 1 - 0 + 1 = T$ (clue at $t = t_e = 0$ and decision-making at $t = t_r = T - 1$). Thus, $\overline{K} = \min_n \{\xi_n\}_n - 1 = T - 1$.

3. By varying the environment parameter $T = L + 1$ or the agent's context size $K$, we can assess the agent's long-term or short-term memory. For instance, if $T$ is fixed, setting $K > \overline{K} = T - 1$ tests short-term memory. To evaluate long-term memory, we must use memory mechanisms $\mu(K)$ and set context length $K \leq \overline{K} = T - 1 \leq K_{eff} = \mu(K)$.

Theoretically, this estimate $K = \overline{K}$ is sufficient to test the long-term memory of an agent, but in practice it is better to choose a value $K$ closer to the left boundary of the interval $[1, \overline{K}]$, as this allows us to track the effect of the memory mechanism $\mu(K)$ more explicitly.

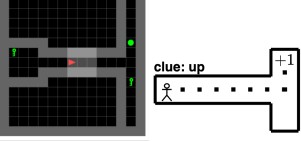

**Minigrid-Memory**  **Passive T-Maze**

Figure 4: Memory-intensive environments for testing short–term memory and long-term memory in Memory DM.

## 6 EXPERIMENTS

To illustrate the importance of following a consistent methodology (Algorithm 1) when evaluating an agent's long-term and short-term memory capabilities, as well as to highlight the ambiguity in results that can arise from experimental misconfigurations, we conducted a series of experiments with memory-enhanced agents in memory-intensive environments using the Memory DM framework.

For our experiments, we chose two memory-intensive environments: 1) Passive-T-Maze (Ni et al., 2023) and 2) Minigrid-Memory (Chevalier-Boisvert et al., 2023) (see Figure 4). In the Passive-T-Maze, the agent starts at the beginning of a T-shaped maze and observes a clue, which it must use to make a turn at a junction at the end of the maze. The Minigrid-Memory environment presents a similar challenge to the Passive-T-Maze; however, the agent must first reach a room containing a clue before walking down a corridor and making a turn. A detailed description of these environments can be found in Appendix E.

As baselines, we chose transformer-based (Deep Transformer Q-Networks (DTQN) (Esslinger et al., 2022), Gated Transformer-XL (GTrXL) (Parisotto et al., 2020), and GPT-2-DQN (Ni et al., 2023)) and RNN-based (Long Short-Term Memory (LSTM) Hochreiter & Schmidhuber (1997)) agents with different memory types according to our classification (see Table 3 and Figure 3).

## 6.1 IMPACT OF EXPERIMENT CONFIGURATION ON MEMORY TYPE TESTED

In this section, we highlight the significance of specifying experiment configurations for testing long-term and short-term memory in memory-intensive environments based on quantitative parameters. We utilized a GTrXL memory-enhanced agent with a long-term memory mechanism in the Minigrid-Memory environment. The configurations included a 31x31 grid with variable corridor lengths (corresponding to $\xi \in [1, \max \Xi)$) and a constant corridor length ($\xi = \max \Xi = \min \Xi$). In the first configuration, the agent was trained with $\xi < K_{eff}$ over multiple episodes, achieving an average reward of $0.95 \pm 0.02$. Here, we tested both short-term memory on a specific set of episodes and long-term memory on the remaining ones, as per the corollary to Theorem 1. In contrast, the second configuration involved training exclusively on episodes where $K_{eff} < \xi = \max \Xi = \min \Xi$, focusing solely on long-term memory, resulting in an average reward of $0.53 \pm 0.04$.

While the GTrXL agent demonstrated some ability to recall events beyond the effective context in the first experiment, the second experiment's results indicated random guessing behavior, suggesting a failure to utilize information outside the effective context. This ambiguity arose because the first experiment did not follow our proposed methodology, inadvertently testing both memory types. In contrast, the second experiment adhered to our methodology ($K_{eff} < \max \Xi = \min \Xi$), allowing for a clear evaluation of the agent's long-term memory capabilities (see Algorithm 1).

## 6.2 TESTING LTM AND STM MEMORY

| Method | Passive-T-Maze | | |
|--------|----------------|---|---|
| | **short-term** ($K = 15, \xi = 15$) | **long-term** ($K = 5, \xi = 15$) | **short-term** ($K = 5, \xi = 5$) |
| GPT-2 DQN | $0.9 \pm 0.1$ | $0.5 \pm 0.2$ | $1.0 \pm 0.0$ |
| DTQN | $1.0 \pm 0.0$ | $0.5 \pm 0.1$ | $1.0 \pm 0.0$ |

Table 2: Performance of models in short-term and long-term memory settings on the Passive-T-Maze

To test the agent's short-term and long-term memory abilities separately, we conducted experiments in two settings according to our proposed methodology. According to the proposed algorithm 1, for Passive T-Maze, we determined a context memory boundary value $\overline{K} = \xi - 1$ and chose agent context length values $K$ such that when testing short-term memory $K > \overline{K}$ and when testing long-term memory $K \leq \overline{K}$.

In the first experiment setting we train model with $K = 15, \xi = 15$ and evaluate them on environment with $K = 5, \xi = 15$. As can be seen from the validation results Table 2, both DTQN (Esslinger et al., 2022) and GPT-2 DQN (Ni et al., 2023) demonstrate short-term memory capabilities and learn perfectly when $K > \xi - 1$. However, when $K \leq \xi - 1$, the models failed to utilize the initial cue provided at the beginning of the game, choosing directions randomly at the end of the maze. This indicates a lack of long-term memory in these models. On the other hand, (GTrXL) (Parisotto et al., 2020) and LSTM DQN (Ni et al., 2023), by definition 1, have a context of $K = 1$, and possess only long-term memory. As seen from the validation results in this setting, this models demonstrate good performance due to the memory mechanisms they incorporate. (GTrXL) (Parisotto et al., 2020) archives mean reward of $1.0 \pm 0.0$ on long-term it uses KV-cache, which allows the model to significantly extend the effective context. The primary memory mechanism in LSTM DQN (Ni et al., 2023) is the recurrent layers, which enable the model to achieve mean reward of $0.9 \pm 0.1$.

Similar to the previous experiment, the agent's long-term and short-term memory can be evaluated in another way: by fixing the model context $K$ and varying the environment parameters. In the second experiment setting, we fixed the context $K = 5$ and trained the models in an environment with $\xi = 5$, followed by validation with $\xi = 15$. The results show a pattern similar to the first experiment (see Table 2, the middle and right columns).

These experiments demonstrates how our Algorithm 1 helps distinguish both short-term and long-term memory capabilities in RL agents.

## 6.3 EXTRAPOLATION TO LARGER CORRELATION HORIZON

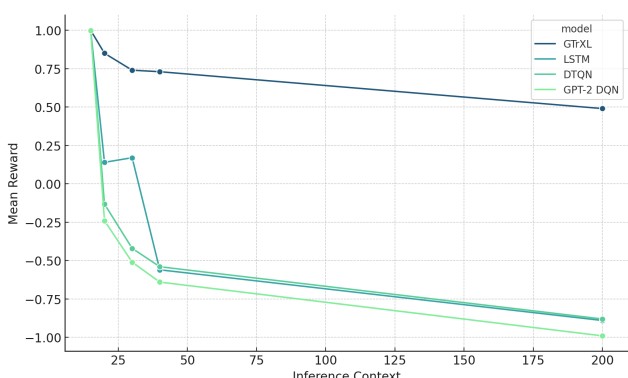

Figure 5: Results of extrapolation abilities for baselines.

In this experiment, we evaluated the ability of models previously trained on the Passive T-Maze to extrapolate to larger correlation horizon. For this, we took checkpoints trained with $\xi = 15$ and conducted inference on longer correlation horizon. The results are shown in Figure 5. As seen from the graph, models that were previously classified as long-term demonstrate the ability to extrapolate to larger contexts, while models with short-term memory experience a significant drop in performance when inferring on contexts longer than those seen during training. This experiment aligns with previous findings and confirms the validity of memory classification in baseline models using the algorithm presented in this paper.

## 7 CONCLUSION

In this study, we propose formal definitions of memory tasks arising in the RL domain, inspired by neuroscience: long-term memory (LTM) vs. short-term memory (STM), declarative memory vs. procedural memory. We also distinguish explicitly two classes of POMDPs: Memory Decision-Making (Memory DM) and Meta-Reinforcement Learning (Meta-RL), survey the main memory mechanisms used in such tasks, partition existing Memory DM algorithms according to our classification, and propose an algorithm for setting up an experiment to test explicitly and correctly an agent LTM and STM capabilities within the Memory DM framework. We have also shown that without following the proposed methodology for conducting an experiment to test LTM and STM, one can obtain ambiguous results that do not allow us to separate these types of agent's memory. We hope that our work will bring clarity to the understanding of the concept of memory in RL and advance progress in the field.

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

## A    APPENDIX – GLOSSARY

In this section, we provide a comprehensive glossary of key terms and concepts used throughout this paper. The definitions are intended to clarify the terminology proposed in our research and to ensure that readers have a clear understanding of the main elements underpinning our work.

1. $\mathcal{M}$ – MDP environment

2. $\mathcal{M}_P$ – POMDP environment

3. $\tilde{\mathcal{M}}_P$ – memory-intensive environment

4. $h_{0:t-1} = \{(o_i, a_i, r_i)\}_{i=0}^{t-1}$ – agent history of interactions with environment

5. $K$ – agent base model context length

6. $\overline{K}$ – context memory border of the agent, such that when $K \in [1, \overline{K}] \rightarrow$ strictly LTM problem

7. $\mu(K)$ – memory mechanism that increases number of steps available to the agent to process

8. $K_{eff} = \mu(K)$ – the agent effective context after applying the memory mechanism

9. $\alpha_{t_e}^{\Delta t} = \{(o_i, a_i, r_i)\}_{i=t_e}^{t_e+\Delta t}$ – an event starting at time $t_e$ and lasting $\Delta t$, which the agent should recall when making a decision in the future

10. $\beta_{t_r} = \beta_{t_r}(\alpha_{t_e}^{\Delta t}) = a_t \mid (o_t, \alpha_{t_e}^{\Delta t})$ – the moment of decision making at time $t_r$ according to the event $\alpha_{t_e}^{\Delta t}$

11. $\xi = t_r - t_a - \Delta t + 1$ – an event's correlation horizon

## B    APPENDIX – MEMORY MECHANISMS

In RL, memory has several meanings, each of which is related to a specific class of different tasks. To solve these tasks, the authors use various memory mechanisms. The most prevalent approach to incorporating memory into an agent is through the use of Recurrent Neural Networks (RNNs) (Rumelhart et al., 1986), which are capable of handling sequential dependencies by maintaining a hidden state that captures information about previous time steps (Wierstra et al., 2010; Hausknecht & Stone, 2015; Sorokin et al., 2015; Duan et al., 2016; Song et al., 2018; Zintgraf et al., 2020). Another popular way to implement memory is to use Transformers (Vaswani et al., 2017), which use self-attention mechanisms to capture dependencies inside the context window (Parisotto et al., 2020; Lampinen et al., 2021; Esslinger et al., 2022; Melo, 2022b; Team et al., 2023; Pramanik et al., 2023; Robine et al., 2023; Ni et al., 2023; Grigsby et al., 2024; Shala et al., 2024). State-space models (SSMs) (Gu et al., 2021; Smith et al., 2023; Gu & Dao, 2023) combine the strengths of RNNs and Transformers and can also serve to implement memory through preservation of system state (Hafner et al., 2019; Lu et al., 2023; Becker et al., 2024; Samsami et al., 2024). Temporal convolutions may be regarded as an effective memory mechanism, whereby information is stored implicitly through the application of learnable filters across the time axis (YuXuan Liu & Hsieh, 2016; Mishra et al., 2018). A world model (Ha & Schmidhuber, 2018) which builds an internal environment representation can also be considered as a form of memory. One method for organizing this internal representation is through the use of a graph, where nodes represent observations within the environment and edges represent actions (Morad et al., 2021; Zhu et al., 2023; Kang et al., 2024b).

A distinct natural realization of memory is the utilization of an external memory buffer, which enables the agent to retrieve pertinent information. This approach can be classified into two categories: read-only (writeless) (Oh et al., 2016; Lampinen et al., 2021; Goyal et al., 2022; Cherepanov et al., 2024) and read/write access (Graves et al., 2016; Zaremba & Sutskever, 2016; Parisotto & Salakhutdinov, 2017). Detailed information about each memory mechanism can be found in the Appendix, Appendix B.

Using these memory mechanisms, both decision-making tasks based on information from the past within a single episode and tasks of fast adaptation to new tasks are solved. However, even in works using the same underlying base architectures to solve the same class of problems, the concepts of memory may differ.

## B.1 RECURRENT NEURAL NETWORKS

Recurrent Neural Networks (RNNs) (Rumelhart et al., 1986) are a class of neural networks designed to process sequential data by maintaining a hidden state that captures information from previous time steps. This hidden state, which is essentially a form of internal memory, allows RNNs to model temporal dependencies and long-range relationships within the input sequence. Variations of RNNs, such as Gated Recurrent Units (GRUs) (Chung et al., 2014) and Long Short-Term Memory (LSTM) (Hochreiter & Schmidhuber, 1997) networks, have been developed to address the vanishing gradient problem and improve the ability to retain long-term information. LSTMs introduce an additional cell state and output gate to better manage the flow of information through time, while GRUs combine the update and reset gates to simplify the LSTM architecture. These enhancements enable RNNs and their variants to effectively handle complex sequential tasks in RL.

RNNs are able to maintain and utilize information from previous states or experiences, which is especially valuable in POMDPs. This capacity is integral for tasks where the current state does not provide sufficient information to make optimal decisions. Prominent RL models that incorporate RNNs include Deep Recurrent Q-Networks (DRQN) (Hausknecht & Stone, 2015) and Deep Attention Recurrent Q-Networks (DARQN) (Sorokin et al., 2015), which extend the capabilities of Deep Q-Networks (DQN) (Mnih et al., 2013) by integrating LSTM layers, and Recurrent Policy Gradient (RPG) (Wierstra et al., 2010) methods, which adapt policy gradient techniques to sequential data. Additionally, Asynchronous Advantage Actor-Critic (A3C) (Mnih et al., 2016) with RNNs and Recurrent Deterministic Policy Gradient (RDPG) (Song et al., 2018) further illustrate the effectiveness of RNNs in complex RL tasks.

In the context of Meta-RL, RNNs are integral to enhancing an agent ability to learn and adapt from temporally correlated data. By maintaining a hidden state that evolves over time, RNNs can effectively encode information about past experiences, which is critical for Meta-RL tasks that require fast adaptation to new environments. The Fast RL via Slow RL ($RL^2$) (Duan et al., 2016) algorithm exemplifies this, utilizing an LSTM network to embed the history of interactions, thereby enabling the agent to quickly infer the optimal policy for a new task. Additionally, the Variational Bayes-Adaptive Deep RL (VariBAD) (Zintgraf et al., 2020) algorithm employs a RNN to maintain a latent representation of the task, which is updated as new information becomes available, thus facilitating rapid adaptation.

## B.2 TRANSFORMERS

Transformers (Vaswani et al., 2017) are a type of neural network architecture designed to process sequential data. Unlike RNNs, which process input sequences sequentially and use recurrence to capture long-range dependencies, Transformers process input sequences in parallel and use self-attention mechanisms to capture dependencies inside the context window. This parallelization allows Transformers to be much faster and more efficient than RNNs, especially for longer sequences.

Transformers have found application in various areas of RL (Li et al., 2023; Agarwal et al., 2023): Online RL (Parisotto et al., 2020; Lampinen et al., 2021; Esslinger et al., 2022; Zheng et al., 2022; Melo, 2022a; Team et al., 2023; Pramanik et al., 2023), Offline RL (Cherepanov et al., 2024; Janner et al., 2021; Lee et al., 2022; Jiang et al., 2023), and model-based RL (Chen et al., 2022; Micheli et al., 2023; Robine et al., 2023), including for solving credit assignment problems and working in memory-intensive environments (Chen et al., 2021; Ni et al., 2023; Grigsby et al., 2024), provided that the entire trajectory fits within the model context. Transformers for Meta-Reinforcement Learning (TrMRL) (Melo, 2022b) is a Meta-RL agent that mimics the memory reinstatement mechanism using the transformer architecture. Hierarchical Transformers for Meta-Reinforcement Learning (HTrMRL) (Shala et al., 2024) encode the same intra-episodic memory as proposed in TrMRL and allow us to understand the transition dynamics within episodes and accesses the intra-episodic memories to encode how the transition dynamics across episodes relate to each other. Gated Transfromer-XL (GTrXL) (Parisotto et al., 2020) uses Transformer-XL (Dai et al., 2019) with GRU-like (Chung et al., 2014) gating mechanism to solve Meta-RL problems.

### B.3 State-space models

State-Space Models (SSMs) are a fundamental concept in control theory, used to describe and analyze dynamic systems by representing the system's behavior through a set of state variables. In the context of deep learning, a subset of SSMs, specifically Linear-Time Invariant (LTI) systems, has gained significant attention due to their remarkable performance and potential as an alternative to traditional transformer-based models. These LTI-based SSMs, exemplified by models like Structured State Space sequence model (S4) (Gu et al., 2021) and implified Structured State Space Sequence Models (S5) (Smith et al., 2023), as well as SSMd with input-dependent matricesMamba (Gu & Dao, 2023), have showcased not only remarkable performance but also computational efficiency, making them a compelling alternative to transformers. To deal with the vanishing/exploding gradients problem (Pascanu et al., 2013) and improve memorization ability these models use high-order polynomial projection operators (HiPPO) theory (Gu et al., 2020) of continuous-time memorization. Their high inference speed and parallelizable training making them potentially useful in RL.

Deep Planning Network (PlaNet) (Hafner et al., 2019) is a Recurrent State Space Model (RSSM) model-based agent that learns the environment dynamics from images and chooses actions through fast online planning in latent space. Modified S5 model (Lu et al., 2023) enables us to initialize and reset the hidden state in parallels used for modeling long-term decision-making tasks and solving Meta-RL problems. KalMamba (Becker et al., 2024) leverages Mamba to learn the dynamics parameters of a linear Gaussian SSM in a latent space. Recall to Image (R2I) (Samsami et al., 2024) combines S5 and Dreamer (Hafner et al., 2020) models.

### B.4 Graphs

A world model is a learned internal representation that captures the agent understanding of the environment. This model acts as a form of memory, enabling the agent to recall past experiences. One way to structure this internal representation is through a graph, where nodes represent observations in the environment and edges represent actions.

In the Graph-Based Memory Reconstruction (GBMR) (Kang et al., 2024b) the memory is implemented as a graph storing states, connections between states, and policy. This graph imitates Bartlett's reconstruction mechanism of human memory. Value Memory Graph (VMG) (Zhu et al., 2023) is a graph-structured world model in Offline-RL setting that represents the original environments as a graph-based MDP. Graph actions are sampled via Value Iteration and then converted to environment actions via the actions translator. Graph Convolutional Memory (GCM) (Morad et al., 2021) receives an observation as input and places it in a graph, determines the neighbors of that vertex and adds edges, and then through graph neural network receives belief state and forms a policy

### B.5 Temporal Convolutions

In the context of neural networks, temporal convolutions function as an efficient memory mechanism for handling sequential data. Unlike RNNs, which use explicit memory cells, Temporal Convolutions implicitly remember information by applying learnable filters across the time axis. This approach enables the network to capture both short-term and long-term dependencies in the data. The convolution operation aggregates features from previous time steps, effectively summarizing historical context and using it to inform predictions about future time steps.

Simple Neural Attentive Meta-Learner (SNAIL) (Mishra et al., 2018) combines Temporal Convolutions to aggregate information from past experience and soft attention to focus on specific pieces of information. A3CTConv is a A3C model (Mnih et al., 2016) augmented with a Temporal Convolution layer after the last convolution layer and A3CTConvRNN is a A3CTConv modification with the recurrent layer (YuXuan Liu & Hsieh, 2016).

### B.6 External storage with read / write operators

Another promising approach to enhancing an agent's memory is to utilize an external information storage with read-only or read/write access, as is common in computer architectures (tapes, RAMs, GPUs, etc.).

External storage with read-only access allows information to be stored sequentially in a buffer, enabling the agent to read data without the ability to modify it. Models employing this type of memory include MemNNs, such as Memory Q-Network (MQN), Recurrent Memory Q-Network (RMQN), and Feedback Recurrent Memory Q-Network (FRMQN) (Oh et al., 2016). Additionally, Retrieval-Augmented Agent (R2A) (Goyal et al., 2022), Hierarchical Chunk Attention Memory (HCAM) (Lampinen et al., 2021), and Episodic Memory Deep Q-Networks (EMDQN) (Lin et al., 2018) exemplify systems that leverage read-only external storage.

In contrast, external storage with read/write access prevents the loss of important information from the buffer and allows for the addition of new data in an addressable manner. This capability enables models to dynamically update their memory based on ongoing experiences. Notable examples of models utilizing this type of memory include Neural Map (Parisotto & Salakhutdinov, 2017), which is designed for structured memory in RL, and Differentiable Neural Computer (DNC) (Graves et al., 2016), which mimics the functionality of traditional computers by separating memory from computation. Reinforcement Learning Neural Turing Machines (RL-NTM) (Zaremba & Sutskever, 2016) also exemplify this approach by incorporating neural networks with external memory capabilities.

Moreover, replay buffers and their variations can also be classified under this type of memory mechanism. Replay buffers are crucial in RL as they store past experiences, allowing agents to learn from a diverse set of previously encountered states. This mechanism helps stabilize learning by breaking the correlation between consecutive experiences, which can lead to more robust training outcomes.

### B.7 Memory in parameters

When RL agents are parameterized by neural networks, the optimization process itself can be seen as a form of memorization process. the simplest example is, feedforward networks that learn to associate outputs with their corresponding inputs. This process updates parameters (or "weights") of neural networks so that each input-output pair is memorized. This type of memory is often reffered to as in-weight memory as a counterpart of in-context memory studied in this work.

From the optimization perspective, certain phenomena or best practices in training neural networks may favour or harm memory when RL agents are implemented by neural networks. These include: the loss of plasticity, exponentially moving average (EMA) teacher in RL, catastrophic forgetting. Loss of plasticity is a phenomenon observed in continual learning when neural network lose their learnability over the sequence of tasks. Catastrophic forgetting is a vulnerability of all modern deep learning models - when faced with a new task, neural network might forget the first task, while it can remember both if it had to solve them simultaneously. Exponentially moving average is a technique, initially proposed to stabilize deep q-networks. It can also be seen as a form of memory since the neural network becomes an average of multiple past versions of it.

Unfortunately, the in-weight memory (as opposed to in-context memory studied in this work) is much less reliable. When agents are tasked to be trained continually, catastrophic forgetting might erase past knowledge and skills learned by RL agent in the series on RL tasks.

### B.8 Memory as world model

Model-based reinforcement learning (MBRL) is another framework which is related to Memory DM. The central idea of MBRL is to train a world model, i.e. the distribution of the form $p_\theta(s_{t+1} \mid s_t, a_t)$ parameterized by trainable parameters $\theta$. In POMDPs , real states are unknown, therefore, world model must learn it from the sequences of actions and observations in an unsupervised or self-supervised fashion. This typically happens through learning of latent states and modelling them sequentially (Hafner et al., 2020; 2022; 2023; Bruce et al., 2024). This mechanism can also be seen as a form of memory – since the world model learns a compressed representation of the environment, it needs to remember events that are relevant for decision making.

## C Appendix – Meta Reinforcement Learning

In this section, we explore the concept of Meta-Reinforcement Learning (Meta-RL), a specialized domain within POMDPs that focuses on equipping agents with the ability to learn from their past

experiences across multiple tasks. This capability is particularly crucial in dynamic environments where agents must adapt quickly to new challenges. By recognizing and memorizing common patterns and structures from previous interactions, agents can enhance their efficiency and effectiveness when facing unseen tasks.

Meta-RL is characterized by the principle of "*learning to learn*", where agents are trained not only to excel at specific tasks but also to generalize their knowledge and rapidly adjust to new tasks with minimal additional training. This adaptability is achieved through a structured approach that involves mapping data collected from various tasks to policies that guide the agent's behavior.

Meta-RL algorithm is a function $f_\theta$ parameterized with *meta-parameters* that maps the data $\mathcal{D}$, obtained during the process of training of RL agent in MDPs (tasks) $\mathcal{M}_i \sim p(\mathcal{M})$, to a policy $\pi_\phi : \phi = f_\theta(\mathcal{D})$. The process of learning the function $f$ is typically referred to as the *outer-loop*, while the resulting function f is called the *inner-loop*. In this context, the parameters $\theta$ are associated with the outer-loop, while the parameters $\phi$ are associated with the inner-loop. Meta-training proceeds by sampling a task from the task distribution, running the inner-loop on it, and optimizing the inner-loop to improve the policies it produces. The interaction of the inner-loop with the task, during which the adaptation happens, is called a *lifetime* or a *trial*. In Meta-RL, it is common for $\mathcal{S}$ and $\mathcal{A}$ to be shared between all of the tasks and the tasks to only differ in the reward $\mathcal{R}(s, a)$ function, the dynamics $\mathcal{P}(s' \mid s, a)$, and initial state distributions $P_0(s_0)$ (Beck et al., 2024). The formal definition of Meta-RL framework is presented in Definition 7.

**Definition 7** (**Meta-RL**). **Meta-RL** – is a class of POMDPs where the agent learns to learn from its past experiences across multiple tasks and memorize the common patterns and structures to facilitate efficient adaptation to new tasks. Let $\mathcal{D} = \{\tau_j^{\mathcal{M}_i}\}_{j=0}^{H-1}$ is all of the data of $H$ episodes of length $T$ collected in the MDP $\mathcal{M}_i \sim p(\mathcal{M})$. A Meta-RL algorithm is a function $f_\theta$ that maps the data $\mathcal{D}$ to a policy $\pi_\phi$, where $\phi = f_\theta(\mathcal{D})$. The objective to determine an optimal $f_\theta$:

$$J^\theta = \mathbb{E}_{\mathcal{M}_i \sim p(\mathcal{M})} \left[ \mathbb{E}_\mathcal{D} \left[ \sum_{\tau \in \mathcal{D}_{I:H}} G_i(\tau) \Big| f_\theta, \mathcal{M}_i \right] \right], \text{ where } G_i(\tau) - \text{discounted return in the MDP } \mathcal{M}_i,$$

$I$ – index of the first episode during the trial in which return counts towards the objective (Beck et al., 2024).

## D   APPENDIX – SURVEY ON MEMORY-AUGMENTED AGENTS IN MEMORY DM FRAMEWORK

To demonstrate the variety of existing approaches for implementing agent memory mechanisms in the Memory DM framework, we have compiled a Table 3 including the main models with each of the considered agent memory types. We also categorized the types of problems into LTM and STM according to our proposed terminology. It is important to note that some of the algorithms presented in Table 3 are designed to use agent memory to solve Meta-RL problems, however, we have included them in the table because these works also performed experiments in the setting we categorized as Memory DM.

In addition to Table 3, Table 4 summarizes the main memory-intensive environments from the related works. As can be seen in Table 4, there are very few environments that have overlap with several different models. This indicates that there is no common set of benchmarks (as, for example, in NLP or CV domains) for Memory DM framework tasks, which makes it difficult to compare the developed algorithms with other baselines.

According subsection 4.2, Meta-RL is a POMDPs class with implicit memory that produce a policy based on the entire agent-environment interaction history. The Definition 4 introduces the concepts of short-term memory and long-term memory in the Memory DM framework. Nevertheless, these concepts can be used to describe the Meta-RL framework when considering POMDP inner-loop tasks. Thus, existing Meta-RL algorithms can be divided into two subclasses: those that use memory mechanisms to transfer skills between tasks and knowledge within tasks (POMDP inner-loop), and those that use memory mechanisms only to transfer skills between tasks (MDP inner-loop).

This work focuses mainly on the Memory DM framework, and therefore we do not provide a survey of Meta-RL algorithms and environments and the algorithm to conduct the experiment to test agent memory. Nevertheless, the terminological framework proposed in this paper allows us to categorize

existing Meta-RL algorithms by memory type and memory mechanisms, which is useful for further research in this area.

Table 3: Classification of RL models with memory in Memory DM framework. $K_{eff}^{tested}$ – the order of the effective context realized in the corresponding work, "Type" – type of tested agent memory, according to our terminology.

| Model | Base model | Online | Memory mechanism $\mu(K)$ | $K_{eff}^{tested}$ | Type |
|---|---|---|---|---|---|
| DRQN (Hausknecht & Stone, 2015) | LSTM | ✓ | RNN | $10^0..10^1$ | LTM |
| DTQN (Esslinger et al., 2022) | Transformer | ✓ | Attention | $10^1$ | STM |
| DARQN (Sorokin et al., 2015) | LSTM | ✓ | RNN | $10^0$ | LTM |
| HCAM (Lampinen et al., 2021) | Transformer | ✓ | Ext. memory w/o write op. | $10^2$ | LTM |
| AMAGO (Grigsby et al., 2024) | Transformer | ✓ | Attention | $10^2..10^4$ | STM |
| RATE (Cherepanov et al., 2024) | Transformer | X | Ext. memory w/o write op. | $10^1..10^2$ | LTM |
| GTrXL (Parisotto et al., 2020) | Transformer | ✓ | Ext. memory w/o write op. | $10^2$ | LTM |
| R2I (Samsami et al., 2024) | SSM | ✓ | SSM + model-based | $10^2..10^3$ | LTM |
| MERLIN (Wayne et al., 2018) | LSTM | ✓ | RNN MBRL + Ext. memory with write op. | $10^1..10^2$ | LTM |
| Modified S5 (Lu et al., 2023) | SSM | ✓ | SSM | $10^1..10^2$ | LTM |
| Neurl Map (Parisotto & Salakhutdinov, 2017) | GRU | ✓ | Ext. memory with write op. | $10^1..10^2$ | LTM |
| GBMR (Kang et al., 2024b) | CNN + FC | ✓ | Graph + model-based | $10^2$ | LTM |
| EMDQN (Lin et al., 2018) | CNN + FC | ✓ | Ext. memory w/o write op. | $10^0$ | LTM |
| MRA (Fortunato et al., 2020) | LSTM | ✓ | RNN | $10^1..10^3$ | LTM |
| FMRQN (Oh et al., 2016) | LSTM + FC | ✓ | RNN + Ext. memory w/o write op. | $10^0..10^1$ | LTM |

Table 4: The main memory-intensive environments used in the reviewed works suitable for testing agent LTM in Memory DM framework. Atari Bellemare et al. (2013) environment is included to demonstrate that many memory-augmented agents are tested exclusively in MDP environments.

| Environment | DRQN (Hausknecht & Stone, 2015) | DTQN (Esslinger et al., 2022) | DARQN (Sorokin et al., 2015) | HCAM (Lampinen et al., 2021) | AMAGO (Grigsby et al., 2024) | RATE (Cherepanov et al., 2024) | GTrXL (Parisotto et al., 2020) | R2I (Samsami et al., 2024) | MERLIN (Wayne et al., 2018) | Modified S5 (Lu et al., 2023) | Neural Map (Parisotto & Salakhutdinov, 2017) | GBMR (Kang et al., 2024b) | EMDQN (Lin et al., 2018) | MRA (Fortunato et al., 2020) | FMRQN (Oh et al., 2016) |
|---|---|---|---|---|---|---|---|---|---|---|---|---|---|---|---|
| Atari (Bellemare et al., 2013) | ✓ | | ✓ | | | ✓ | | ✓ | | | | ✓ | ✓ | | |
| T-Maze (Ni et al., 2023) | | | | | ✓ | ✓ | | | | | | | | | |
| ViZDoom-Two-Colors (Sorokin et al., 2022) | | | | | | ✓ | | | | | | | | | |
| Minigrid.Memory (Chevalier-Boisvert et al., 2023) | | | | | | ✓ | | | | | | | | | |
| Memory Maze (Pasukonis et al., 2022) | | | | | | ✓ | | ✓ | | | | | | | |
| HeavenHell (Geffner & Bonet, 1998) | | ✓ | | | | | | | | | | | | | |
| Hallway (Littman et al., 1995) | | ✓ | | | | | | | | | | | | | |
| Car Flag (Nguyen, 2021) | | ✓ | | | | | | | | | | | | | |
| Gym-Gridverse (Baisero & Katt, 2021) | | ✓ | | | | | | | | | | | | | |
| DMLab-30 (Beattie et al., 2016) | | | | | | | ✓ | | | ✓ | | | | | |
| POPGym (Morad et al., 2023) | | | | | | ✓ | | ✓ | | ✓ | | | | | |
| BSuite (Osband et al., 2019) | | | | | | | | ✓ | | | | | | | |
| Ballet (Lampinen et al., 2021) | | | | ✓ | | | | | | | | | | | |
| Object permanence (Lampinen et al., 2021) | | | | ✓ | | | | | | | | | | | |
| Memory Game (Wayne et al., 2018) | | | | | | | | | ✓ | | | | | | |
| Goal-Search (Parisotto & Salakhutdinov, 2017) | | | | | | | | | | | ✓ | | | | |
| PsychLab (Leibo et al., 2018) | | | | | | | | | | | | | | ✓ | |
| Spot the Difference (Fortunato et al., 2020) | | | | | | | | | | | | | | ✓ | |
| Navigate to Goal (Fortunato et al., 2020) | | | | | | | | | | | | | | ✓ | |
| I-Maze (Oh et al., 2016) | | | | | | | | | | | | | | | ✓ |
| Pattern Matching (Oh et al., 2016) | | | | | | | | | | | | | | | ✓ |

# E   APPENDIX – ENVIRONMENTS DESCRIPTION

This section provides an extended description of the environments used in this work.

**Passive-T-Maze (Ni et al., 2023).**   In this T-shaped maze environment, the agent's goal is to move from the starting point to the junction and make the correct turn based on an initial signal. The agent can select from four possible actions: $a \in left, up, right, down$. The signal, denoted by the

variable $clue$, is provided only at the beginning of the trajectory and indicates whether the agent should turn up ($clue = 1$) or down ($clue = -1$). The episode duration is constrained to $T = L + 1$, where $L$ is the length of the corridor leading to the junction, which adds complexity to the task. To facilitate navigation, a binary variable called $flag$ is included in the observation vector. This variable equals $1$ one step before reaching the junction and $0$ at all other times, indicating the agent's proximity to the junction. Additionally, a noise channel introduces random integer values from the set $-1, 0, +1$ into the observation vector, further complicating the task. The observation vector is defined as $o = [y, clue, flag, noise]$, where $y$ represents the vertical coordinate.

The agent receives a reward only at the end of the episode, which depends on whether it makes a correct turn at the junction. A correct turn yields a reward of $1$, while an incorrect turn results in a reward of $0$. This configuration differs from the conventional Passive T-Maze environment (Ni et al., 2023) by featuring distinct observations and reward structures, thereby presenting a more intricate set of conditions for the agent to navigate and learn within a defined time constraint. To transition from a sparse reward function to a dense reward function, the environment is parameterized using a penalty defined as $penalty = -\frac{1}{T-1}$, which imposes a penalty on the agent for each step taken within the environment. Thus, this environment has a 1D vector space of observations, a discrete action space, and sparse and dense configurations of the reward function.

**Minigrid-Memory (Chevalier-Boisvert et al., 2023).** Minigrid-Memory is a two-dimensional grid-based environment specifically crafted to evaluate an agent's long-term memory and credit assignment capabilities. The layout consists of a T-shaped maze featuring a small room at the corridor's outset, which contains an object. The agent is instantiated at a random position within the corridor. Its objective is to navigate to the chamber, observe and memorize the object, then proceed to the junction at the maze's terminus and turn towards the direction where the object, identical to that in the initial chamber, is situated. A reward function defined as $r = 1 - 0.9 \times \frac{t}{T}$ is awarded upon successful completion, while failure results in a reward of zero. The episode concludes when the agent either makes a turn at a junction or exhausts a predefined time limit of $95$ steps. To implement partial observability, observational constraints are imposed on the agent, limiting its view to a $3 \times 3$ frame size. Thus, this environment has a 2D space of image observations, a discrete action space, and sparse reward function.

