# OpenReview forum: "Unraveling the Complexity of Memory in RL Agents: an Approach for Classification and Evaluation"
_ICLR.cc/2025/Conference — Submitted to ICLR 2025_

### Official Review · Reviewer_QTL2 · 2024-11-01

**Soundness:** 3
**Presentation:** 2
**Contribution:** 2
**Rating:** 5
**Confidence:** 3

**Summary:**

The paper introduces an approach inspired by human cognitive abilities to formalise memory types in reinforcement learning (RL), providing precise definitions for short-term memory (STM) and long-term memory (LTM). STM is defined as the agent's reliance on recent interactions, while LTM involves recalling information over longer time intervals outside of the immediate context. The authors differentiate between Meta-Reinforcement Learning (Meta-RL), which focuses on cross-task skill transfer (procedural memory), and Memory Decision-Making (Memory DM), where agents use historical data within a single environment (declarative memory).

In the Memory DM setting, the authors develop a rigorous evaluation methodology to assess memory capabilities in RL agents. This approach is validated in memory-intensive environments, such as the Passive T-Maze and Minigrid-Memory, by varying critical parameters—context length (the memory span an agent can handle) and correlation horizon (the temporal dependency between events). By varying key parameters, the experiments demonstrate the Memory DM framework’s ability to reliably assess STM and LTM in RL agents.

**Strengths:**

* The paper provides neuroscience-based definitions of memory types, clarifying RL memory research, which enables more accurate agent comparisons and tailored evaluation methods for each type​. The cognitive science-inspired approach has interdisciplinary appeal, likely to attract interest from both RL and cognitive science researchers, fostering potential collaboration and cross-disciplinary insights.
* The paper’s methodology is grounded in theoretical rigour, offering a scientifically robust framework that enhances the validity and reliability of memory evaluation in RL studies.
* It introduces a standardised methodology for assessing memory capabilities, promoting reproducibility and consistency across RL studies by providing clear criteria for experimental setups.

**Weaknesses:**

* The framework has been validated in simple environments, which may not capture the challenges of more sophisticated settings or real-world scenarios, potentially limiting its practical applicability.
* The paper discusses procedural memory as part of its classification scheme but does not provide or suggest an evaluation methodology related to it, focusing solely on declarative memory. This results in an incomplete validation and leaves open questions about the classification’s practical application to skill-transfer scenarios.
* The methodology section is dense and complex, additional visual aids or examples could clarify the experimental design and enhance comprehension for a broader audience.

**Questions:**

* Could the framework be extended to evaluate procedural memory in Meta-RL settings? Are there specific experiments that could be added to address skill transfer across tasks?
* In my interpretation, declarative and procedural memory are intended as distinct concepts; however, the definitions in Equation 2 of Definition 3 imply that declarative memory could be included within procedural memory due to the “or” condition and “≥” allow for overlap. Could the authors clarify whether declarative memory is meant to be a subset of procedural memory or fully distinct? How does this impact the proposed distinction between Memory DM and Meta-RL in the framework?
* How does this framework compare to existing memory evaluation approaches in RL? What are the specific advantages of using cognitive science-inspired definitions over more traditional RL memory metrics?
* What motivates the specific classification of memory types (declarative vs. procedural, STM vs. LTM), and how does it improve memory assessment in RL over a general approach?

---

> ### Author Response · Authors · 2024-11-23
> **Response to Reviewer QTL2**
>
> Thank you for highlighting the strengths of our work. We respond to the comments below.
>
> **W1. Simple environments**
>
> On one hand, the environments may seem relatively simple; however, they enable rapid and targeted exploration of various aspects of memory and hypothesis testing. In addition, the T-Maze environment is a common standard memory test in RL [1, 2, 3].
>
> The goal of these environments is to facilitate testing specific hypotheses related to memory without requiring the RL agent to solve additional tasks that involve learning unrelated skills or abilities.These examples are enough to demonstrate the core ideas of the paper.
>
> **W2, Q1. Procedural memory**
>
> Definitely, testing procedural memory is an important and valuable task; however, it lies beyond the scope of our work. We provide a definition of procedural memory along with examples to distinguish it from declarative memory. In this study, we focus primarily on declarative memory and propose a method for testing it.
>
> **W3. Visual aids or examples**
>
> We have aimed to support all key concepts introduced in the article with examples and visualizations. For instance, the illustrations in Fig. 1, Fig. 2, and Fig. 3 visualize and help clarify the definitions of declarative and procedural memory, long-term and short-term memory, and the general classification of memory types.
>
> We are confident that, in addition to formal definitions, these visualizations will assist readers in better understanding the concepts and terms introduced in the paper, making the work more accessible to a wider audience.
>
> **Q2: Declarative vs. procedural memory**
>
> Declarative memory and procedural memory are indeed defined in the article as conceptually distinct concepts. Declarative memory is described as the use of knowledge within a single episode and a single environment, whereas procedural memory encompasses skill transfer across multiple episodes or environments. We agree that the current wording may lead to misinterpretation, and therefore, we have adjusted the definition to make it more precise:
>
>
> - Declarative Memory  $\Leftrightarrow (n_{envs}\times n_{eps}=1)$
>
> - Procedural Memory $\Leftrightarrow (n_{envs}\times n_{eps}>1)$
>
> **Q3, Q4**
>
> **Advantages of cognitive science definitions**
>
> Definitions from cognitive science, such as short-term and long-term memory, as well as declarative and procedural memory, are already well-established in the RL community, but do not have common meanings and are interpreted in different ways. We strictly formalize these definitions to avoid possible confusion that may arise when introducing new concepts.
>
> **Memory classification**
>
> The motivation for introducing a classification of different types of memory with respect to temporal dependencies and types of memorized information is motivated by practical goals. In the course of our research on memory in RL and a review of existing work in this area, we concluded that modern challenges and tasks in RL require such a classification to ensure proper memory testing in RL agents.
>
> For instance, some interpret memory as employing transformers with extensive context windows, others as utilizing recurrent networks, and still others as a model’s ability to transfer skills across tasks. However, these approaches often differ fundamentally in design, making direct comparisons under identical conditions potentially invalid, or testing conditions suitable for one agent may not align with another’s memory mechanism.
>
> **How does this framework compare to existing memory evaluation approaches in RL?**
>
> Currently, in RL memory mechanisms are tested by running agents in memory-intensive environments and evaluating metrics without considering environment or agent temporal configurations. Section 6.1 shows the issues with this approach.
>
> This experiment shows how naive validation of an agent with memory in a memory-intensive environment can lead to incorrect conclusions about its memory type. We derive three configurations of context length K and correlation horizons $\xi$ from Theorem 1 to evaluate: 1) long-term memory, 2) both long- and short-term memory, and 3) long-term memory only.
>
> Our results demonstrate that the same agent trained in the same environment produces different results based on the K and $\xi$ configuration. When using our proposed Algorithm 1, the agent learns 0.53 ± 0.04, indicating its inability to solve the long-term problem. However, with the default configuration, it achieves 0.95 ± 0.02, which might suggest long-term memory, but since both long- and short-term memory were tested, we cannot definitively claim it has long-term memory.
>
> [1] Esslinger, K. et al.. Deep transformer q-networks for partially observable reinforcement learning. arXiv:2206.01078.
>
> [2] Grigsby, J. et al. Amago: Scalable in-context reinforcement learning for adaptive agents.ICLR 2024
>
> [3] Pramanik, S. et al. AGaLiTe: Approximate Gated Linear Transformers for Online Reinforcement Learning.TMLR 2024

---

> > ### Comment · Reviewer_QTL2 · 2024-11-25
> >
> > I appreciate your thoughtful response to the weaknesses I raised and your flexibility in updating the paper based on my review. However, about Q2, I am still confused regarding your reply. You mentioned that "Declarative memory and procedural memory are indeed defined in the article as conceptually distinct concepts" and indicated an update to $n_{\text{envs}} \times n_{\text{eps}} > 1$. Yet, in the paper, I found Procedural Memory defined as $n_{\text{envs}} \times n_{\text{eps}} \geq 1$.
> >
> > Regarding the advantages, I agree with the points raised in your reply, the need to formalise distinctions between memory types to address inconsistencies in RL research and ensure standardised testing for memory mechanisms. These efforts provide a valuable conceptual framework and contribute significantly to advancing the field, however, while I recognise these strengths, the innovation and motivation as presented in the paper still fall short of convincing me to raise my scores.

---

> > > ### Author Response · Authors · 2024-11-25
> > >
> > > Thank you for your reply!
> > >
> > > Regarding Q2, - the updated definition from our response here is actual, i.e. with a “>” sign. We thank you for helping us to correct this typo in the text of the paper.
> > >
> > > We would be very grateful if you could tell us what exactly you are questioning about our work after the response we have provided. It is very important for us to get this feedback in order to clearly communicate our ideas to the community.

---

> ### Comment · Reviewer_QTL2 · 2024-11-29
>
> Thank you for addressing the typo in the paper. Related to my remaining concerns about novelty primarily related to the scope of the work. While procedural memory is intentionally left outside the experimental scope, its absence from evaluation feels like a missed opportunity especially that the flow of the paper initially introduces Memory DM and  Meta-RL, but the testing methodology focuses solely on Memory DM, leaving the discussion incomplete.

---

> > ### Author Response · Authors · 2024-11-29
> >
> > We appreciate your valuable feedback and the contribution to improving the paper.
> >
> > The terminology introduced in our work aims to allow us to explicitly distinguish between types of agent memory. Through this separation, we distinguish declarative memory, which we focus on in the following, and for which we propose a validation methodology for STM and LTM.
> >
> > We do not consider the Meta-RL framework because we initially started working on memory formalization because the tasks we defined as Memory DM were mixed with Meta-RL, being fundamentally different in nature. By abstracting away from Meta-RL, we were able to provide a methodology for evaluating STM and LTM in the Memory DM framework clearly.
> >
> > Moreover, as noted in Table 1, Meta-RL with POMDP inner-loop tasks and Meta-RL with MDP inner-loop tasks are also fundamentally different tasks that should be evaluated in a different way, and this evaluation is out-of-scope of our work, as we are solely interested in declarative memory.
> >
> > We hope that our answer was able to clarify your concerns and will have a positive impact on your evaluation of our work.

---

### Official Review · Reviewer_weyF · 2024-11-03

**Soundness:** 3
**Presentation:** 3
**Contribution:** 3
**Rating:** 5
**Confidence:** 4

**Summary:**

The paper attempts to create clarity in the use of the term "memory" in a reinforcement learning context.  As well as suggesting definitions for different kinds of memory and different memory related tasks, the authors present a more rigorous way for testing memory capabilities of reinforcement learning techniques and show possible pitfalls of violating the proposed methodology.

**Strengths:**

I really like the paper and the topic it presents.  I think it is good to have a more clear definition of what exactly is meant by memory and what its contribution can be in reinforcement learning.  I like the approach the authors came up with and the clarity with which they presented it.  I think there is a need for a paper like this, and I like how the authors looked at the current state of affairs in reinforcement learning research and its treatment of a new and important branch in the field that deals with memory in a bunch of different contexts.

**Weaknesses:**

However, the topic seems difficult to deal with in a conference paper.  When reading the introduction and the goal of the paper as set out, I was expecting a more broad overview of current use of the memory term and the different ways it is used and abused in reinforcement learning literature and research.  I think the topic is very interesting, but a paper doing a deep dive into a topic such as this has to build a clear foundation for its contributions by taking the body of existing work into account (To be clear, I am not suggesting that the authors don't do this.) and illustrating this by giving a broad overview of said existing work in the paper.

In my view, this contribution wants to be presented in a review paper, with an overview of recent existing work laying a strong foundation for the contributions made by the authors, namely, bringing clarity to the current mismatch in use of the term "memory" in the field.

Currently, the paper includes a very brief section on POMDPs, which are important, but don't represent all ways in which the term memory is used.  However, since this is section 2, I think this is a bit misleading, as it seem to set the context in full.  The related works section is very brief, and much related work is relegated to the appendix, where most of it is only referenced, but not placed in context of the suggested structure and definitions.  Section 4 lays some foundation from cognitive science and RL, and talks about the credit assignment problem in relation to memory handling, but it feels rushed and the role or importance it plays isn't obvious.  All of this should be given more room to be elaborated on.  I understand that this is impossible in a conference paper however, but now it feels rushed, and I don't feel the work will get the attention it deserves or reach the audience that it should.

**Questions:**

This is where double blind reviewing sucks, as I would strongly recommend that the authors produce the review paper, with additional contribution this paper could be, and I would happily serve as a reviewer for said paper.

---

> ### Author Response · Authors · 2024-11-23
> **Response to Reviewer weyF**
>
> We appreciate the reviewer's comments and are glad to hear that you share our vision of the need to formalize concepts related to memory in RL. We respond to the reviewer’s comments below.
>
> **Broad overview of use of the memory term**
>
> Our work is not a review paper. Our main goal is to formalize the basic concepts of memory in RL so that new memory-enhanced agents in RL can be compared and validated correctly. We take the definitions of memory from neuroscience as the basis of our formalism, since they have been used in RL for a long time, albeit in different senses, as we write about in Section 3 - Related Works. In turn, in Appendix B - Memory Mechanisms, we give an extensive description of what is basically meant by memory in RL.
>
> **Abuse of the term “memory”**
>
> We cannot talk about the abuse of the term “memory” precisely because everyone understands memory differently. For example, someone understands memory as the use of a transformer with a very large context window, someone understands it as the use of recurrent networks, and someone understands it in general as the ability of a model to transfer skills from one task to another. At the same time, these algorithms may have fundamental differences in their design and comparing them under the same conditions may not be correct, or the conditions for testing memory mechanisms for one agent may not be appropriate for another.
>
> That is why we offer our definitions of different types of memory that allow us to fully describe the basic concepts in RL that are commonly understood by memory.
>
> **Purely review article**
>
> Writing a purely review article would be interesting to us as well, and we'll get into that next. However, for the moment, our goal is to propose practical ways to separate agent memory types in RL, as well as their validation in the Memory DM framework.
>
> **POMDPs**
>
> The definitions we propose are based on the POMDPs formalism in both the Memory DM and Meta-RL contexts, which is why we have placed this section at the beginning.
>
> **Related Works**
>
> In the Related Works section, since this is not a review paper but a practical one, we show that RL actively uses memory types from neuroscience, but with different meanings. Rather than providing a comprehensive review of all works on memory mechanisms in RL, we focus on our proposed contributions. Consequently, we do not conduct a detailed classification of existing memory mechanisms; instead, we provide an overview of them in the Appendix.
>
> **Cognitive science and RL**
>
> We base our definitions of memory on neuroscience, as concepts from neuroscience have long been used in Memory RL, as we discuss in the Related Works Section. A deeper look into cognitive science aspects of memory is out-of-scope for our work, as we focus specifically on the practical application of our taxonomy and memory validation algorithm.
>
> Thank you for your interest in our work and your valuable feedback. We appreciate your proposal to serve as a reviewer, which encourages us to deepen our research. We look forward to discussing your ideas and suggestions to improve the paper and hope for further collaboration to enhance its quality.

---

> > ### Comment · Reviewer_weyF · 2024-11-28
> >
> > Thank you for your reply.  I don't disagree with you about the fact that you didn't write a review paper but instead attempt to formalize the concept of memory within RL.  However, to get the community behind you and agree with your formalization, I think a review might be required to place your suggestion in context.  I might be wrong about this, but suggested this to enhance your chances of impact.
> >
> > I am sorry about using the expression "use and abuse".  I didn't mean RL people are using it incorrectly, but I wanted to emphasize the difference in how people use the term and think about it, just like you stated in your answer.

---

> > > ### Author Response · Authors · 2024-11-28
> > >
> > > Thank you for your response! We appreciate that you have clarified your position on our work.
> > >
> > > In our reply we have tried to answer all your concerns and we hope that we have been able to clarify the positioning of our work. Please, tell us, do you have any other comments/suggestions about our work? We would be very happy to continue the discussion.

---

### Official Review · Reviewer_nevP · 2024-11-11

**Soundness:** 2
**Presentation:** 2
**Contribution:** 2
**Rating:** 3
**Confidence:** 4

**Summary:**

This is a sort of conceptual paper, it's main concern is to taxonomize the concept of memory in reinforcement learning. Given the taxonomy, it aims to demonstrate why paying attention to the categories it suggests is important for interpreting the results of experiments on RL agents involving memory.

**Strengths:**

I'm generally favorably disposed toward conceptual papers like this. And I do agree with the authors that their target, memory in RL, is a worthy target for such an effort.

The main distinction is between declarative and procedural first, and then short-term versus long-term second. The latter is defined with respect to a context length parameter. I do think it's a good idea to highlight somehow the difference between associations within the context window and outside of it. This is a very relevant difference with many algorithmic implications.

**Weaknesses:**

- This is trying to be a conceptual paper aimed squarely in the intersection between AI and cognitive /neuro science. However, judged in that way, I don’t think it really makes the grade. The problem is that it doesn’t really connect clearly into the conversation on the cognitive / neuro science side. There are very few references to these disciplines for one, or less than I would expect any way. And critical references for multiple memory systems are missing (there are so many, but I like some of Squire's old papers on the topic).  And there is basically no context in the paper connecting the work to the ways that researchers in these other fields have thought about memory. For a paper like this which purports to offer a formalization of what is meant by ‘memory’, it's clearly important to relate the new formalization to old ones and discuss how they are similar and different, and to try to sustain an argument for why the present one is an advance on the old.

- I’m not buying the claim that RL is capturing declarative memory. I would tend to say that a defining feature of declarative memory as opposed to procedural memory is the declarative memory’s arbitrariness. The classic prototype example of a declarative memory is a person’s name. And most definitions you find look something like "declarative memory is defined as the type of memory that involves consciously recalling facts and events". It’s all very language-like. But RL memories aren’t usually like that. In some cases they may be, but it’s not too common memory in RL outside of language data. I would probably have been more forgiving of this claim to capture declarative memory with RL had come a few years ago, but now that we have LLMs, what’s the point in trying to get all bent out of shape to capture declarative memory in RL? The prototype of declarative memory is arbitrary information conveyed by language e.g. “My teacher’s name is Bob” (episodic flavor) or “Paris is the capital of France” (semantic flavor). So it seems very reasonable to expect a model of declarative memory to use the kinds of AI systems that work for that kind of data, now that they exist and are so widespread. And of course there are plenty of ways to combine RL and LLMs. I understand though that that would take this too far afield for the present work. And this isn’t really a computational neuroscience modeling paper. So this isn’t really a weakness of the paper here. No need to reply to this bullet point since I don’t think it really matters for your paper. But I’m just leaving it here as a way to convey a bit more my mindset with regard to this paper. At very least, the arbitrariness of the associations seems like a critical part of declarative memory.

- Right after positing a difference between declarative and procedural memory in terms of the algorithms that implement them, in the very next paragraph it then acts like this distinction is established and ready to support further claims when it says “many studies fail to differentiate between agents with declarative and procedural memory”. But that’s  not a strong argument given that it just followed right after defining these terms. Why should other papers have tried to probe things according to the arbitrary categories you just defined?  Especially since, I suspect many researchers would not necessarily agree with the classification. At any rate, the paper merely asserts that one set of tasks are declarative and another are procedural, but it offers no evidence that this distinction corresponds to what others mean by those terms.

- Definition 3 says one should call RL problems involving a “single environment”  problems of declarative memory and RL problems featuring multiple environments problems of procedural memory. This definition would be  impossible to apply in practice. It would appear to suggest that all memory is declarative since one can always compose “multiple environments” together into a single meta-environment. The difference between one environment or many environments is not in the task itself, it’s just a purely formal aspect of the modeling language. One generally is not supposed to predicate a general definition on such a purely formal property since it would make your classifications float around following specific and contingent task parameterization properties.  Note: all the same comments also apply to the episode concept too.

**Questions:**

1. In this paper, is memory meant to be a problem or a solution?

2. Is there some kind of Marr-Poggio levels of analysis story that could be used to clarify the overall structure of the argument here?

3. Doesn't this taxonomy seem to bundle together too many things that really are not similar? Replay buffers used just for training and dynamically accessed external memories, used at test time, are quite different algorithmically, and used in quite different ways. Why does this classification scheme seem to drop these in the same bucket? (There is text in the appendix suggesting this). Also, I don't see why it's even true according to the definition in the main text. I would think that definition would separate these approaches.  And that would. be kind of the whole point of separating declarative and procedural memory. Why doesn't it separate them in the appendix anymore?

4. The paper includes the following sentence in the results section “This ambiguity arose because the first experiment did not follow our proposed methodology”, well this certainly doesn’t inspire any confidence. Why talk about an experiment that doesn’t fit the proposed methodology? It's likely I've misunderstood this paragraph. I find it very hard to follow this part.

5. How would you think about an RL model that remembers and reproduces a time interval? E.g. [Deverett, B., et al. (2019). Interval timing in deep reinforcement learning agents. NeurIPS.]  That paper showed that purely feedforward agents can sometimes solve what appear to be memory tasks. Does that matter? How would your definitions classify the feedforward agent in that paper?

---

> ### Author Response · Authors · 2024-11-23
> **Response to Reviewer nevP**
>
> We appreciate that the reviewer feedback and. Here are our responses.
>
> **W1. Focusing on RL**
>
> Our work focuses exclusively on RL, using neuroscience concepts as reference points to define memory types. We use the neuroscience framework because the terms “long-term/short-term, working, episodic memory, etc.” are already used in RL, but without a unified meaning. Therefore, we redefine them with clear, quantitative meanings to specify the type of agent memory, since the performance of many algorithms depends on their type of memory. We do not claim to have enumerated all types of human memory, since our work is focused exclusively on RL.
>
> **W2. RL is capturing declarative memory**
>
> Declarative memory of RL agents involves recalling and reusing arbitrary “facts,” which can be any environmental representation (e.g., a door’s color) learned during training and do not necessarily have to be verbalized. Our work focuses on Memory DM tasks, where agents use historical data for decisions. While LLMs excel at language tasks, declarative memory in RL often involves signal processing, making its study a distinct challenge.
>
> **W3. Declarative and procedural memory in RL**
>
> In RL, “memory” is used similarly in Meta-RL and Memory DM tasks, despite requiring different memory types, leading to comparison and validation issues (of two algorithms with a stated long-term memory one may not solve the same simple problems as the other one). To address this, we define procedural and declarative memory to clarify their roles in specific tasks.
>
> **W4. Practical application**
>
> Our definitions of declarative and procedural memory in RL are practical, such that they use two numerical metrics: the number of environments n_{envs} and episodes n_{eps}, enabling clear identification of the memory type required for a task. An environment refers to where the agent interacts and receives feedback, while an episode is the sequence from start to terminal state. These concepts are rooted in RL and widely used in existing benchmarks and baselines.
>
> **Q1. Memory meant to be a problem or a solution?**
>
> Memory can be both a problem and a solution, ​​which is why we titled our paper this way. In POMDPs, it helps agents overcome the challenge of incomplete environmental information by storing and retrieving past interactions. However, implementing an effective memory mechanism is a complex challenge.
>
> **Q2. Marr-Poggio levels**
>
> Yes, we consider that the methodology we propose for testing long/short-term declarative memory can be described using Marr-Poggio levels of analysis. However, this requires disclosure of par. 4 “Analyze the results,” in Algorithm 1, to formalize the system’s outputs. In the current version, we leave this point to the independent interpretation of researchers.
>
> - Computational
>
> Goal: Test long/short-term declarative memory in Memory DM tasks
> Input: Memory model, Oracle agent acting random at memory event recall, memory-intensive environment (Theorem 1).
> Language: Standard POMDP/MDP formalism, along with memory definitions from our paper.
> Output: 1 if memory model has tested type of memory, and 0 otherwise
>
> - Algorithmic
>
> Use Algorithm 1 to test long/short-term memory.
>
> - Implementation
>
> Focus on implementing memory-intensive environments and memory mechanisms, though the methodology is not tied to specific implementations.
>
> **Q3. Replay buffer**
>
> In Appendix we give examples of different memory mechanisms as they are understood in various works in the RL domain: “'In RL, memory has several meanings, each of which is related to a specific class of different tasks”.
>
> We added replay buffers to the text due to the fact that well-known works [1, 2] treat it as a memory mechanism.
>
> Q4. “Experiment did not follow methodology”.
>
> This experiment shows how naive validation of an agent with memory in a memory-intensive environment can lead to incorrect conclusions about its memory type. We derive three configurations of context length K and correlation horizons $\xi$ from Theorem 1 to evaluate: 1) long-term memory, 2) both long- and short-term memory, and 3) long-term memory only.
>
> Our results demonstrate that the same agent trained in the same environment produces different results based on the K and $\xi$ configuration. When using our proposed Algorithm 1, the agent learns 0.53 ± 0.04, indicating its inability to solve the long-term problem. However, with the default configuration, it achieves 0.95 ± 0.02, which might suggest long-term memory, but since both long- and short-term memory were tested, we cannot definitively claim it has long-term memory.
>
> **Q5. Time interval RL**
>
> In accordance with the definitions we introduced, in article [3], a feedforward agent employs a long-term declarative memory based on autostigmergy [3] to solve the task of reproducing time intervals.

---

> ### Author Response · Authors · 2024-11-23
>
> [1] Mnih, V. (2013). Playing atari with deep reinforcement learning. arXiv preprint arXiv:1312.5602.
>
> [2] Schaul, T. (2015). Prioritized Experience Replay. arXiv preprint arXiv:1511.05952.
>
> [3] Deverett B. (2019). Interval timing in deep reinforcement learning agents //NeurIPS 2019

---

> > ### Comment · Reviewer_nevP · 2024-11-24
> >
> > Thank you for your detailed response. However I am still not convinced. The response doubles down on some of the points I disagree with in the paper. In particular, I don't think it's helpful for the community to have computer science papers  redefining neuroscience terms in ways that cause them to diverge so much from the way they are used in neuroscience. And especially to do so in a way that responds only to one particularistic framework (reinforcement learning in this case), and to do so with little reference to the literature that created the terminology, the reasons they did so, and ignoring the characteristic phenomena usually considered to fit in each category.
> >
> > I'll leave my score as it is.

---

> > > ### Author Response · Authors · 2024-11-25
> > > **RL is not a particularistic framework, it's our target area**
> > >
> > > Thank you for your reply!
> > >
> > > You think that **we should not redefine terms from neuroscience to computer science**, and we agree with you, because **that is what our paper is about**. In RL, which is the focus of our work, the terms used in our paper are already well-established, so we are not the first to use them. We share your point of view regarding the divergence of meanings in neuroscience and RL, and in order to prevent these divergences from continuing to confuse the understanding of what memory is in RL, we propose to anchor certain meanings in a specific way that researchers in RL talk about in their papers.
> > >
> > > We have **highlighted the main concepts** that are understood by memory in RL and r**elated them to similar concepts from neuroscience** that have already entered the field of RL. Thus, using our definitions, it is not necessary to introduce new entities.
> > >
> > > We do not claim to have complete definitions for all sciences, as we **consider RL exclusively**. We are confident that our proposed definitions bring clarity to the field of Memory RL and will further contribute to an even more active development of this field.
> > >
> > > We hope to continue the discussion on this topic, as we find your comments very valuable for our work.

---

### Author Response · Authors · 2024-12-04
**General Response**

We are very grateful to the reviewers for their detailed and valuable feedback on our work. We are especially grateful that they highlighted key strengths, including our conceptual approach (Reviewer nevP), the emphasis on distinguishing agent memory types in RL (Reviewers nevP, weyF, and QTL2), and the rigorous formalism of our methodology (Reviewer QTL2).

Our work is not aimed at the neurosciences and the precise treatment of memory in them. Instead, we use memory concepts from the neurosciences with respect to entities in the Memory RL domain that have a similar meaning. We use just such definitions from the neurosciences because they are already well-established in the Memory RL domain but are still interpreted differently in different works. The lack of common meanings for the same definitions leads to the possibility that agents with memory may be compared incorrectly, or memory mechanisms may be validated or used incorrectly.

Our proposed terminology allows us to correct this ambiguity and, based on quantitative characteristics, accurately characterize the type of memory needed for a particular task. Additionally, our proposed algorithm for validating agent memory in the Memory DM framework allows us to explicitly separate and evaluate both short-term memory and long-term memory.

We believe that we have answered all of the reviewers' questions and comments and hope that this will have a positive impact on our final evaluation.

---

### Meta-Review · Area_Chair_zsZy · 2024-12-20

**Metareview:**

This was a conceptual paper which proposed an RL-based way of thinking about memory with some arguable connection to neuroscience, unfortunately, reviewers did not find the correspondence to neuroscience convincing or adequately referenced. There were also issues with experimental evaluations.

**Additional Comments On Reviewer Discussion:**

Reviewers engaged in the discussion but did not want to increase their ratings.

---

### Decision · Program_Chairs · 2025-01-22

Reject